# LiDAR Positioning Algorithm Based on ICP and Artificial Landmarks Assistance

**DOI:** 10.3390/s21217141

**Published:** 2021-10-28

**Authors:** Qingxi Zeng, Yuchao Kan, Xiaodong Tao, Yixuan Hu

**Affiliations:** 1College of Automation Engineering, Nanjing University of Aeronautics and Astronautics, Nanjing 211106, China; jslyzqx@nuaa.edu.cn (Q.Z.); taoxiaodong1999@163.com (X.T.); hyxuanly@163.com (Y.H.); 2Nondestructive Detection and Monitoring Technology for High Speed Transportation Facilities, Key Laboratory of Ministry of Industry and Information Technology, Nanjing University of Aeronautics and Astronautics, Nanjing 211106, China; 3The Key Laboratory of Nuclear Technology Application and Radiation Protection in Astronautics, Ministry of Industry and Information Technology, Nanjing University of Aeronautics and Astronautics, Nanjing 211106, China; 4State Key Laboratory for Automobile Simulation and Control, Jilin University, Jilin 130015, China

**Keywords:** landmarks detection, ICP with landmarks, reflector matching

## Abstract

As one of the automated guided vehicle (AGV) positioning methods, the LiDAR positioning method, based on artificial landmarks, has been widely used in warehousing logistics industries in recent years. However, the traditional LiDAR positioning method based on artificial landmarks mainly depends on the three-point positioning method, the performance of which is limited due to landmarks’ layout and detection requirements. This paper proposes a LiDAR positioning algorithm based on iterative closest point (ICP) and artificial landmarks assistance. It provides improvements based on the traditional ICP algorithm. The result of positioning provided by the landmarks is used as the initial iteration ICP value. The combination of the ICP algorithm and landmarks enables the positioning algorithm to maintain a certain positioning precision when landmark detection is disturbed. By comparing the proposed algorithm with the positioning scheme developed by SICK in Germany, we prove that the combination of the ICP algorithm and landmarks can effectively improve the robustness under the premise of ensuring precision.

## 1. Introduction

The positioning method based on LiDAR is one of the leading technologies applied to AGV positioning. According to VDI 2510-2005 [1], AGV is a kind of autonomous vehicle running on the ground, automatically controlled and guided by the non-contact guidance system. At present, the electromagnetic AGV positioning technology is more mature, while the positioning system based on LiDAR and vision is gradually applied to AGV with research developments [2]. Electromagnetic positioning [3,4] requires sensors to get the signal of the electromagnetic wire set on the ground in advance to obtain the deviation between AGV and the preset track. The limitations of the visual systems are mainly related to the calculation cost and light and imaging conditions (i.e., direct sunlight, shadows, image blur, image scale changes, etc.). They are more easily affected by hardware and environment [5]. Besides, other technologies such as inertial navigation system (INS) [6], ultra wide band (UWB) [7] and radio frequency identification (RFID) [8] are also applied to AGV positioning but they also have limitations. The LiDAR positioning systems are more flexible and can realize the accurate positioning in most environments without the track being preset in advance [9]. At present, the LiDAR positioning systems are usually based on artificial landmarks and natural features. LiDAR positioning systems based on artificial landmarks rely on reflectors with high reflection intensity and the three-point positioning method [10]. By matching the reflectors detected at the current position to the ones in the global map stored in advance, the global coordinates of reflectors can be obtained. If there are at least three obtained global coordinates, the current global coordinates of the LiDAR can be calculated by the three-point positioning method. LiDAR positioning systems based on natural features usually rely on simultaneous localization and mapping (SLAM) [11], including 2D SLAM such as GMapping [12] and Cartographer [13], as well as 3D SLAM like LiDAR odometry and mapping (LOAM) [14]. Considering the cost of 2D LiDAR and 3D LiDAR, LiDAR positioning systems based on 2D SLAM are more widely used. LiDAR positioning systems based on the SLAM algorithm suit scenes with plentiful geometric features, especially corner features. In contrast, scenes with inadequate geometric features such as corridors and warehouses limit the performance. The LiDAR positioning systems based on artificial landmarks have higher precision without the requirements of the environment. However, there is a specific restriction on the layout of artificial landmarks in the scene that LiDAR must detect at least three reflectors at any time. The two methods have their advantages and disadvantages.

Regarding the restriction on the layout of artificial landmarks, Xu et al. [15] proposed a new scheme to calculate the correct position when the number of detected reflectors is less than three. The paper discussed three cases. The global coordinates of AGV are judged according to the intersection point between the circle formed by two or one reflector and the route of AGV, which is known in advance. When there is no reflector detected, the biaxial accelerometer is used to achieve a relatively accurate positioning in a short time based on a known route. The scheme shows that the worst precision is better than 8 cm and is more robust than a single LiDAR positioning system. However, the limitation of this method is that the route of AGV must be known in advance, which affects the flexibility of the LiDAR positioning system. Morrison et al. [16] proposed an algorithm that spurns the reflectors and uses geometric shapes as landmarks. The algorithm extracts geometric features from LiDAR data and matches the features to geometric models such as cylinder, cone and sphere, which can effectively reduce data. The use of natural features can well solve the problem of reflector detection in AGV positioning. However, like the LiDAR positioning system based on natural features, it is required that there are sufficient geometric features to ensure the implementation of the algorithm. This will limit the performance of the algorithm.

In addition, the speed of the reflector matching will decrease when the number of reflectors increases. This will affect the positioning systems’ real-time performance. Therefore, many scholars have proposed new reflector matching algorithms to improve real-time performance. Xu et al. [17] designed a reflector matching method based on dead reckoning. The method predicts the position and orientation of AGV by using an encoder and gyroscope and calculated the possible position of reflectors according to the prediction results. According to the matching of reflectors’ predicted values and observed values, a fast reflector matching method is realized, ensuring accurate matching of the AGV reflector under static and dynamic conditions. However, the article conducted short-distance experimental verification. The performance of this algorithm for long distances is unknown.

Regarding improving positioning precision, Ronzoni et al. [18] proposed an outlier elimination algorithm for the possible matching problem due to measurement error and false detection caused by the reflecting surfaces in the environment. Besides, a reflector matching method was proposed according to the distance between reflectors, which can complete the reflector matching in 0.3 s. However, this algorithm was not applied to practical positioning, and the performance remains unknown.

A high-precision, low-cost AGV integrated navigation method was proposed by Xu et al. [19]. The method fused the dead reckoning with the LiDAR positioning data by the extended Kalman filter. Based on the fusion, the pose is estimated under the minimum mean square error. Due to LiDAR positioning, noise cannot be accurately evaluated; the precision of EKF filtering will decrease. Thus, Xu et al. designed an on-line algorithm based on fuzzy PID to adjust observation noise covariance. The on-line algorithm performed well in innovation covariance estimation. When the given observation noise is not consistent with the actual noise, the algorithm adjusts the observation noise covariance to the actual level quickly. This ensures high precision in filtering. While the algorithm has some engineering reference value, it has not been verified in practice.

In addition to the positioning systems based on reflectors, scholars have gradually applied SLAM technology to AGV positioning. In the traditional EKF algorithm, the error from the bearing variance will accumulate without a closed-loop route guiding the AGV based on SLAM. To solve this, Ye et al. [20] proposed a new SLAM method based on EKF. The method built a map with the help of an electronic compass. The compass provides absolute bearing information, which makes the map more accurate with a simple open loop. The EKF algorithm includes two parts. The inner one uses the encoder information to compute the position and pose of AGV. Then it corrects the estimated result based on the actual data of the compass. Finally, the outer layer of EKF uses the result to correct and map with LiDAR data by further estimating the position and pose of AGV. The RMS error shown in the experiment was 0.0091 m. however, the experiment was conducted under the condition of AGV driven 14.3 m at 0.1 m/s speed; this cannot show the performance well.

Cho et al. [21] presented an AGV SLAM based on the matching method and extended Kalman filter (EKF) SLAM for semiconductor factories where landmarks are unavailable. Firstly, a matching method between the four points in the established global map and the local four points is used to calculate the similar position of the real global position. Secondly, the ICP matching method based on the corrected position is used to estimate the correct position. Finally, the LiDAR measurement error is compensated by EKF. The experimental results show that the average error of the algorithm is 3.6 cm. But the algorithm needs a global map with enough natural features in advance. Once the map changes, the positioning precision will be affected. Moreover, the localization precision can be low due to sensor error, incorrect matching error, and disturbance. There is a high possibility that an error is large during rotational driving.

According to the present problems of AGV LiDAR positioning methods, this paper improves the traditional ICP algorithm and proposes an AGV positioning method based on the ICP algorithm and assisted by artificial landmarks. The main contributions of this paper are as follows: (1) A radius-constrained reflector coordinates fitting method is proposed, which does not require iterative calculation, and the error is less than one centimeter. (2) The iterative initial value of the ICP algorithm is provided by the reflector positioning method. Compared with the traditional reflector positioning method, the matching time decreases and the possibility of wrong reflector matching is reduced. (3) By combining natural and artificial features, not only are the advantages of high precision retained, but also the robustness is improved so that the system can maintain good positioning precision when the number of reflectors is less than three.

The remaining content of this paper is organized as follows. Section 2 describes the steps of the reflector coordinates fitting algorithm in detail. The experiments of the proposed method are shown in Section 3. Section 4 discusses the effectiveness of the method in this paper through multiple experiments. Finally, the conclusion of this paper is presented in Section 5.

## 2. Materials and Methods

### 2.1. Traditional Reflector Positioning Algorithm and Traditional ICP Algorithm

As described in Section 1, the traditional reflector positioning system is more precise; this advantage has led to the system being widely used. However, the limitation of detecting three reflectors at any time is fatal. Conversely, positioning systems based on natural features are more flexible. Therefore, combining the two systems can effectively use their advantages. Their principles are as follows.

#### 2.1.1. Traditional Reflector Positioning Algorithm

The traditional reflector positioning algorithm is a global positioning method. The required hardware is shown in Figure 1, including some reflectors indicated by dotted lines and a single-line LiDAR indicated by a solid line.

The specific process of the traditional reflector positioning algorithm is shown in Figure 2.

First, the reflectors are laid in the scene where AGVs run under the premise of ensuring that LiDAR can detect at least three reflectors at any time. LiDAR scans the reflectors in the current position. For the next position, LiDAR still scans some reflectors and finds the same reflectors in the two groups of reflectors. The same reflectors are used to calculate the pose transformation of the two positions. The newly detected reflectors in the next position are added to the global map based on the pose transformation. If all reflectors in the environment are added, the global map is successfully constructed.

Second, LiDAR moves in the environment and scans reflectors. If the number of detected reflectors is greater than two, the distances of all detected reflector pairs are calculated. Then, in the global map, a group of reflectors, in which the distances of all reflector pairs uniquely correspond to the calculated distances, is found. Based on the correspondences, the local and global coordinates of detected reflectors are obtained.

Whether in the local or global coordinate system, the distance between LiDAR and a reflector is constant. Therefore, the local coordinates of detected reflectors can be used to calculate the distances between LiDAR and reflectors in the local coordinate system. Finally, a global coordinate of LiDAR is found that makes distances between LiDAR and reflectors in the global coordinate system the same as the distances calculated in the local coordinate system.

#### 2.1.2. Traditional ICP Algorithm

The ICP algorithm is a classic positioning method based on natural features. It is also used in AGV positioning. Compared with SLAM, the ICP algorithm requires less computing resources. However, ICP is mostly used to assist other positioning algorithms [22], such as speeding up matching reflectors in a global map. The ICP algorithm can also be used alone for positioning [23,24]. The specific ICP algorithm process is shown in Figure 3.

The purpose of the ICP algorithm is to find the overlap of environmental information described by two given point cloud sets $P^{\prime }$ and $X^{\prime }$. Let $P=\;\left\{p_{1},p_{2}\ldots p_{n}\right\}$ and $X=\;\left\{x_{1},x_{2}\ldots x_{n}\right\}$ represent the overlap in P and *X* where $p_{i}$ and $x_{i}$ are 2D or 3D coordinates as point pairs marked by blue lines in Figure 4a.

Then the rotation matrix R and translation matrix t, which make the P coincide with X after rotation and translation, are calculated according to P and X. However, due to measurement error, matrices R and matrix t do not exist. Therefore, the purpose of ICP changes to minimize the error function Equation (1) with a rotation matrix R and a translation matrix t.
(1)$$E\left(R,t\right)\;=\overset{n}{\underset{i=1}{\sum }}{\left|\left|x_{i}-\;\left(Rp_{i}+t\right)\right|\right|}^{2}$$

The correspondence of overlap points in $P^{\prime }$ and $X^{\prime }$ is unknown in advance. However, due to the high scanning frequency of LiDAR, the pose transformation between consecutive point cloud frames is not too large. Therefore, it is reasonable to regard $x_{n}$, which is the closest point of $p_{n}$ in $X^{\prime }$, as the correspondence of $p_{n}$, as shown in Figure 4a. When all the point pairs are searched with the restriction of a certain threshold, the rotation matrix R and translation matrix t that minimize Equation (1) are calculated by singular value decomposition (SVD) [25]. The specific SVD process is as follows: At first, the centers of P and X are calculated by Equation (2).
(2)$$\begin{array}{c} u_{p}=\frac{1}{n}\overset{n}{\underset{1}{\sum }}p_{i}\\ u_{x}=\frac{1}{n}\overset{n}{\underset{1}{\sum }}x_{i} \end{array}$$

Then the covariance matrix of P and X are calculated by Equation (3).
(3)$$W_{px}=\overset{n}{\underset{1}{\sum }}\left(p_{i}-u_{p}\right){\left(x_{i}-u_{x}\right)}^{T}$$

The covariance matrix $W_{px}$ by SVD is decomposed, as shown in Equation (4).
(4)$$W_{px}=U\left[\begin{array}{ccc} \sigma_{1} & \cdots & 0\\ \vdots & \ddots & \vdots \\ 0 & \cdots & \sigma_{n} \end{array}\right]V^{T}$$
where $\sigma_{1}\cdots \sigma_{n}$ represent the singular values of $W_{px}$, the columns of matrix U are the eigenvectors of $W_{px}W_{px}{}^{T}$, and the columns of matrix V are the eigenvectors of $W_{px}{}^{T}W_{px}$. Then, the rotation matrix R and translation matrix t can be obtained by Equation (5).
(5)$$R=VU^{T}\;t=u_{x}-Ru_{p}$$

Here, R and t are calculated based on the closest point correspondences of $P^{\prime }$ and $X^{\prime }$. This method of finding correspondence is reasonable but not completely accurate. However, the new P=RP+t is closer to X than the initial P. Thus, the closest points in the new P and X are chosen as correspondence again and the new R and t are calculated as shown in Figure 4b,c. The process of finding correspondence calculating R and t is repeated when the error function in Equation (1) is less than a threshold or when the number of iterations reaches a threshold. Finally, the rotation matrix R and translation matrix t, which represent the pose transformation between two point sets, are obtained.

### 2.2. ICP Fused with Reflector Positioning

The traditional ICP algorithm performs well in scenes with plentiful geometric features but has nonideal precision in environments such as corridors and warehouses. Due to iterations in ICP, the real-time performance of ICP is limited without an appropriate initial iteration value provided by hardware such as IMU or an encoder. Even the iteration will converge to a local optimal value, which will negatively impact the positioning. The reflector positioning system can add features to the environment and provide a relatively accurate initial iteration value for the ICP algorithm. This can reduce the number of iterations and prevent convergence on a local optimal value. By adding reflectors as features to the environment, the restrictions on the use environment are reduced.

#### 2.2.1. Coordinates of Reflector Fitting Based on Least Square

To integrate reflector positioning into the ICP algorithm, obtaining accurate reflector coordinates is necessary. An example of a reflector is shown in Figure 5a. A reflector is a cylinder with a length of 1 m and a diameter of 75 mm covered by a reflective membrane based on microprism retroreflector technology [26] on the lower half. Microprism retroreflector technology makes laser beams’ intensities that radiate on the membranes larger than those that radiate on normal materials. The dimensions of reflectors are not constant: They depend on LiDAR’s scanning range and resolution, as shown in Figure 5b. The radius of a reflector must ensure that LiDAR can scan enough reflector point clouds within a certain range to fit the coordinates of reflectors. The relationship between number of reflector point clouds N, scanning range L, resolution R, and radius r is
$$\frac{arctan\frac{r}{l}}{R}\geq \frac{N}{2}.$$

In this paper, N is 5, L is 7 m, and R is 0.15°; r is determined to be greater than 0.0375 m by calculation. The length of a reflector should ensure that the reflective membrane can be scanned.

According to this property, the point clouds of reflectors can be distinguished from other point clouds. The shape of the cylinder makes the reflector show the same circle from any scanning angle. It is much easier to fit a certain shape than an uncertain shape. Therefor, the problem of fitting coordinates of reflectors can be described as fitting the center of a circle with a known diameter based on the point clouds distinguished from other point clouds according to intensities.

Figure 6a shows a vertical view of a reflector placed close to a wall. Figure 6b shows the point clouds $S=\;\left\{S_{i}\left(x_{i},y_{i},f_{i}\right)|i\in \;\left\{1,2,\ldots ,n\right\}\right\}$ obtained by LiDAR scanning the reflector and part of the wall. $\left(x_{i},y_{i}\right)$ and $f_{i}$ represent the coordinate and intensity of the ith reflector, respectively. Set a threshold $f_{t}$; the point clouds whose $f_{i}$ is bigger than $f_{t}$ are the reflector’s point clouds, such as the red points shown in Figure 6b.

Let $R=\;\left\{R_{k}\left(x_{Rk},y_{Rk}\right)|k\in \;\left\{1,2,\ldots ,m\right\}\right\}$ be the set of reflector points. $\left(x_{Rk},y_{Rk}\right)$ represents the coordinate of the kth point cloud of the reflector. By minimizing the error function Equation (6), the coordinate of the reflector can be fitted based on the least square method [27] as shown in Equation (7).
(6)$$E\left(x_{c},y_{c},r\right)\;=\sum_{1}^{m}{\left(x_{Rk}-x_{c}\right)}^{2}+{\left(y_{Rk}-y_{c}\right)}^{2}-r^{2}{)}^{2}$$
(7)$$\begin{array}{c} C=m\sum x_{Rk}^{2}-\sum x_{Rk}\sum x_{Rk}\\ D=m\sum x_{Rk}y_{Rk}-\sum x_{Rk}\sum y_{Rk}\\ E=m\sum x_{Rk}^{3}+m\sum x_{Rk}y_{Rk}^{2}-\sum \left(x_{Rk}^{2}+y_{Rk}^{2}\right)\sum x_{Rk}\\ F=m\sum y_{Rk}^{2}-\sum y_{Rk}\sum y_{Rk}\\ G=m\sum x_{Rk}^{2}y_{Rk}+m\sum y_{Rk}^{3}-\sum \left(x_{Rk}^{2}+y_{Rk}^{2}\right)\sum y_{Rk}\\ a=\frac{GD\;-\;EF}{CF\;-\;D^{2}}\\ b=\frac{GD\;-\;ED}{D^{2}\;-\;CF}\\ c=-\frac{\sum \left(x_{Rk}^{2}\;+\;y_{Rk}^{2}\right)\;+\;a\sum x_{Rk}\;+\;b\sum y_{Rk}}{m}\\ x_{c}=-\frac{a}{2}\\ y_{c}=-\frac{b}{2}\\ r=\frac{\sqrt{a^{2}\;+\;b^{2}\;-\;4c}}{2} \end{array}$$
where r is the radius of the reflector, $x_{c}$ is the the abscissa, and $y_{c}$ is the ordinate of the circle’s center. C to G and a to c represent the value of formulas to simplify the final result without practical meaning. The performance of the above traditional least square is shown in Figure 7.

By comparing Figure 6a and Figure 7, the circle fitted by the traditional least square is not close enough to the point clouds of the wall: The reflector is close to the wall, as shown in Figure 6a. The distance between the reflector’s center and the wall should be equal to the reflector’s radius. Therefore, the fitting precision remains to be improved. It is reasonable to modify the coordinate with known radius and the surrounding points of the reflector. Based on this, the specific implementation steps of modifying the fitted coordinate are summarized as follows: The first step is removing scan shadows [28,29]. The shape of the facula on the object onto which an ideal laser beam radiates is a point. However, a laser beam has a certain divergence angle, which makes the facula an area. Because of this, when there is an object in front of the other, the laser beam may radiate on the edge of the front object. The facula splits into two parts: One part is on the front object, and the other is on the rear object. This makes the reflected laser beam a superposition of the two parts of faculae. LiDAR then determines that the target is between the two objects and result is the phenomenon called scan shadow, shown as blue points in Figure 8.

Such scan shadows affect fitting the wall’s position. Due to their cause, scan shadows occur along the side of the object’s edge. They form vectors with the reflector’s center. Here, the angle at which a vector rotates counterclockwise is defined as a positive value. Based on this, ∠BCA is greater than zero and ∠DCB is less than zero. This means vector $\vec{CA}$ is parallel to vector $\vec{CB}$ after rotating a certain number of degrees counterclockwise. Similarly, vector $\vec{CB}$ is parallel to vector $\vec{CD}$ after rotating a certain number of degrees clockwise. Point B is the end point of scan shadows and the rotation direction changes at this point. Therefore, scan shadows can be judged by adjacent vectors’ rotation direction. As shown in Figure 8, points A and B are the scan shadows to be removed. The vectors formed by surrounding reflector points and the circle’s center are described as in Equation (8).
(8)$$\begin{array}{c} \vec{CA}=\;\left(x_{A}-x_{c},y_{A}-y_{c}\right)\\ \vec{CB}=\;\left(x_{B}-x_{c},y_{B}-y_{c}\right)\\ \vec{CD}=\;\left(x_{D}-x_{c},y_{D}-y_{c}\right) \end{array}$$

Then angles of adjacent vectors are calculated by Equation (9).
(9)$$\begin{array}{c} ∠BCA=arctan\left(\frac{y_{B}\;-\;y_{c}}{x_{B}\;-\;x_{c}}\right)\;-\;arctan\left(\frac{y_{A}\;-\;y_{c}}{x_{A}\;-\;x_{c}}\right)\\ ∠DCB=arctan\left(\frac{y_{D}\;-\;y_{c}}{x_{D}\;-\;x_{c}}\right)\;-\;arctan\left(\frac{y_{B}\;-\;y_{c}}{x_{B}\;-\;x_{c}}\right) \end{array}$$

If the three adjacent points are scan shadows, the angles between them are both positive or negative. If not, one of the angles is positive and the other is negative. Such a property can be represented by multiplication. If ∠BCA×∠DCB is greater than zero, it means ∠BCA and ∠DCB are both positive or negative. When ∠BCA×∠DCB is less than zero, the sign before the two angles is different and point B is the end point of the scan shadows. By such a method, the end of the scan shadow can be determined. The beginning is the closest point of the reflector points, such as point A in Figure 8. Scan shadows can be removed according to their beginning and end.

After removing the scan shadows, the second step is determining the boundaries of the wall. As shown in Figure 6a, the wall to which the reflector is close is a flat plane. Therefore, the cast of the wall in point clouds is a straight line. Before fitting the line, the boundaries must first be determined. Two kinds of boundaries are presented in Figure 9.

The first kind is called the edge point and the other is called the corner point. The edge point is defined as the edge of the wall, such as point $P_{E}$ in Figure 9. On the edge of the wall, the ranges of the edge point cloud and its next point cloud that do not belong to the wall could undergo an acute change. The standard of determining the edge point is based on this property. The change between the ranges of two consecutive laser beams $CP_{E}$ and $CP_{W}$ is defined as Equation (10).
(10)$$\xi =\frac{\left|CP_{E}-CP_{W}\right|}{max\left(CP_{E},CP_{W}\right)}$$

The geometric meaning expressed in Equation (10) is the ratio of the difference between the two ranges’ values to the larger one. The molecule represents the absolute value of the range difference. Due to the property of similar triangles, the difference between the two ranges increases when the distance between LiDAR and the wall increases. Therefore, the range difference between the two laser beams needs to be divided into the range. In other words, the absolute value of change is replaced by the relative rate of change. When the rate of change is greater than the set threshold, this point can be determined as the edge point, such as point $P_{E}$ in Figure 9.

In addition to the edge point, the corner point is the other kind of boundary. Just as the name implies, the corner point is the point cloud representing the wall’s corner. It can be defined by the curvature shown in Equation (11) from reference [30].
(11)$$c=\frac{1}{\left|S\right|\;\cdot \;\|X_{\left(k,i\right)}^{L}\|}\|\underset{j\in S,j\neq i}{\sum }\left(X_{\left(k,i\right)}^{L}-X_{\left(k,j\right)}^{L}\right)\|$$
where $\left|S\right|$ represents the number of points surrounding the point whose curvature is being calculated; $\left\|X_{\left(k,i\right)}^{L}\right\|$ represents the distance between the point and LiDAR, namely the range of the point cloud; $\left\|\underset{j\in S,j\neq i}{\sum }\left(X_{\left(k,i\right)}^{L}-X_{\left(k,j\right)}^{L}\right)\right\|$ is the sum of vectors formed by the surrounding $\left|S\right|$ points and the calculated point. When a point’s curvature is greater than the set threshold and the largest of all curvatures, this point can be determined as a corner point. The two kinds of boundaries can ensure the shape of the wall is a straight line and easy to fit.

The set $W=\;\left\{W_{k}\left(x_{k},y_{k}\right)|k\in \;\left\{1,2,\ldots ,w\right\}\right\}$ of wall points can be determined by traversing a certain number of points adjacent to scan shadows on both sides to find the boundaries. Then the third step is fitting a line of the wall by the least square method [31], as shown in Equation (12).
(12)$$\begin{array}{c} X=\;\left[\begin{array}{cc} x_{1} & 1\\ \vdots & \vdots \\ x_{w} & 1 \end{array}\right]\\ Y=\;\left[\begin{array}{c} y_{1}\\ \vdots \\ y_{w} \end{array}\right]\\ A=\;\left[\begin{array}{c} k\\ b \end{array}\right]\;=\;{\left(X^{T}X\right)}^{-1}X^{T}Y\\ y=kx+b \end{array}$$
where y=kx+b is the function of the line. The result of fitting is shown in Figure 10.

The last step is modifying the initial coordinate of the circle’s center. The radius of the circle is known and the reflector is placed close to the wall. Therefore, the distance between the center and the wall equals the radius. According to this, the method of modifying involves moving the initial coordinate to a point whose distance from the wall is the radius.

As shown in Figure 11, the problem of moving the center equals calculating the length of CC′ and the direction of vector $\vec{CQ}$. The length of CH using Equation (13) is calculated first.
(13)$$CH=\frac{\left|kA-B+b\right|}{\sqrt{k^{2}+1}}$$

CQ is a straight line through the origin and center whose function is $y=\frac{y_{c}}{x_{c}}x$. As the intersection point of line CQ and the line of the wall, the coordinate of point Q is $\left(\frac{b}{\frac{y_{c}}{x_{c}}-k},\frac{y_{c}b}{y_{c}-kx_{c}}\right)$. Then the length of CQ can be obtained based on the coordinates of points C and Q. The distance between the modified center and the wall equals the radius; the length of ${HH}^{\prime }$ equals radius r. ${\Delta H}^{\prime }{CC}^{\prime }\;∼\;\Delta HCQ$ is known because CH is perpendicular to HQ. Because of the properties of similar triangles, $\frac{{CC}^{\prime }}{{CH}^{\prime }}=\frac{CQ}{CH}$, namely ${CC}^{\prime }={CH}^{\prime }*\frac{CQ}{CH}$. Now, the distance that the center needs to be moved is obtained. Finally, the displacements on the X-axis and Y-axis and the modified coordinate are calculated by Equation (14).
(14)$$\begin{array}{c} dis\_x={CC}^{\prime }*\frac{x_{c}}{\sqrt{x_{c}^{2}\;+\;y_{c}^{2}}}\\ dis\_y={CC}^{\prime }*\frac{y_{c}}{\sqrt{x_{c}^{2}\;+\;y_{c}^{2}}}\\ \;\;x_{c^{\prime }}=x_{c}+dis\_x\\ \;\;y_{c^{\prime }}=y_{c}+dis\_y \end{array}$$

The final result of modification is shown in Figure 12 and the precision of fitting the coordinate of reflectors is shown in Section 5.

#### 2.2.2. Improved Reflector Positioning Method

After obtaining all the coordinates of reflectors of each point cloud frame, the pose transformation between frames can be calculated according to the coordinates of corresponding reflectors between adjacent frames. This provides a more accurate iterative initial value for the ICP algorithm.

The correspondence between the reflectors needs to be determined first to obtain the pose transformation. As shown in Figure 13, the symbols of the reflectors in two consecutive point cloud frames scanned by LiDAR in two positions are ABCFGH and BCDEFG. The four reflectors BCFG included in both frames can be used for positioning. The shape and size of each reflector are the same and there is no sign for identification. Therefore, it is necessary to use some features between reflectors for matching the reflector between two frames. A reflector can be considered as a point; two points in a plane can form a line segment. Similar to the traditional positioning system, the line segments formed by reflectors are selected as the matching standard. However, if only line segments are selected as the matching standard, mismatch can easily occur, such as line AH and line DE in Figure 13. Because their lengths are similar, they are easy to successfully match. Then, false reflector pairs AD and HE are constructed.

For reducing such mismatches, the match standard should be three line segments instead of a single line segment. In other words, the triangles formed by reflectors in two frames must be matched line segment by line segment. Only can matching following this method be considered successful. The specific steps are as follows: The first step is calculating all the triangles. The set of triangles in coordinate system XOY of the first frame is shown as
(15)$$Tri=\;\left\{hij,hik,hil,hig,hjk,hjl,hjg,hkl,hkg,hlg,ijk,ijl,ijg,ikl,ikg,ilg,jkl,jkg,jlg,klg\right\}$$

Due to measurement error, three theoretically collinear reflectors such as BCD and EFG can still form a triangle. The set of triangles in coordinate system $X^{\prime }O^{\prime }Y^{\prime }$ of the second frame is shown as: (16)$${Tri}^{\prime }=\{bcd,bce,bcf,bca,bde,bdf,bda,bef,bea,\;bfa,cde,cdf,cda,cef,cea,cfa,def,dea,dfa,efa\}$$

Every triangle symbol in these two sets stores the lengths of the three lines formed by the three reflectors. Constructing the sets starts from the closest reflector such as h and c. The reason is that the speed of AGV limits the pose transformation of point clouds in two adjacent frames. The closest point in the second frame has the highest possibility of corresponding to the first frame. Therefore, constructing triangle sets starting from the closest point can accelerate the matching process.

After obtaining the sets of triangles in the first and second frames, the second step is finding the correspondence of the reflectors. The matching method compares the three lengths of every triangle in ${Tri}^{\prime }$ and the three lengths of every triangle in Tri. For example, the reflectors *BCF* form triangle bcd in ${Tri}^{\prime }$ and triangle hkg in Tri. First, compare three line pairs bc and hk, cd and kg, and bd and gh. The lines are shown in Figure 13. This match obviously fails because the lengths of line pairs are not similar. Then, compare the following three line pairs constructed in another order bc and kg, cd and hg, and bd and hk. This match also fails. Lastly, compare the three line pairs constructed in order bc and gh, cd and hk, and bd and kg. This time, the match is successful. In terms of theory, there are six orders of line pairs. However, the triangles of ${Tri}^{\prime }$ and Tri are all constructed in the same order, clockwise or counterclockwise. For example, triangles bcd and hkg are constructed with reflectors *BCF* counterclockwise. Therefore, the two triangles only need to be matched at most three times.

If triangles are matched successfully, three reflector pairs b and g, c and h, and d and k are obtained. When the first match is successful, a set Pairs of reflector pairs is constructed. If the newly obtained reflector pairs do not exist in Pairs, add them to Pairs and set their scores to one. If they exist, take the scores of the existing reflector pairs plus one. For example, when bcd and hkg match successfully, the set Pairs is constructed as $Pairs=\;\left\{\left(b,g,1\right),\left(c,h,1\right),\left(d,k,1\right)\right\}$. For the second triangle bce in ${Tri}^{\prime }$, it will match successfully with hlg in Tri. Thus, three reflector pairs bg, ch, and el are obtained. However, in Pairs, pairs bg and ch exist while pair el does not exist. Therefore, the scores of pairs bg and ch plus one and pair el are added to Pairs. The initial set $Pairs=\;\left\{\left(b,g,1\right),\left(c,h,1\right),\left(d,k,1\right)\right\}$ becomes a new set $Pairs=\;\left\{\left(b,g,2\right),\left(c,h,2\right),\left(d,k,1\right),\left(e,l,1\right)\right\}$. As such, when all the triangles including reflector *b* in ${Tri}^{\prime }$ are matched, a set of reflector pairs is obtained $Pairs=\{\left(b,g,4\right),\left(c,h,4\right),\left(d,k,4\right),\left(e,l,4\right),\left(b,j,1\right),\left(c,k,1\right),\left(d,h,1\right))\}$. If all the triangles containing the closest reflector in ${Tri}^{\prime }$ are matched unsuccessfully, then match all remaining triangles to find the correspondence of the reflectors.

While the set of reflector pairs includes reflector pairs, nothing can guarantee that all pairs are correct. For example, reflectors BCF construct triangle bcd in ${Tri}^{\prime }$ and reflectors ECF construct triangle hjk in Tri. The two triangles have similar lengths of three lines. Three reflector pairs bj, ck, and dh are obtained with the above matching method. However, reflector b in ${Tri}^{\prime }$ is matched with reflector g in Tri, while mismatched with reflector *j* in Tri. Under any condition, one reflector in ${Tri}^{\prime }$ can only match one reflector in Tri. Therefore, the scores of reflector pairs are used to remove mismatched reflector pairs. In the set Pairs, the pair $\left(b,g\right)$ score is four while that of pair $\left(b,j\right)$ is one. Due to this, the matched triangles bcd in ${Tri}^{\prime }$ and hjk in Tri are considered mismatched and are removed. Finally, the reflector pair set Pairs becomes ${Pairs}^{\prime }=\;\left\{\left(b,g,4\right),\left(c,h,4\right),\left(d,k,4\right),\left(e,l,4\right)\right\}$ after removing mismatched triangles.

After the previous steps, correct reflector pairs are included in ${Pairs}^{\prime }$. Reflectors can be seen as points, so reflector pairs are the correspondences of points as well. The condition that the correspondences of points are known to calculate the pose transformation is similar to that of the ICP algorithm. Similarly, calculate the pose transformation by SV. ${Pairs}^{\prime }$ can be divided into two sets $P=\;\left\{P_{1}\left(x_{b},y_{b}\right),P_{2}\left(x_{c},y_{c}\right),P_{3}\left(x_{d},y_{d}\right),P_{4}\left(x_{e},y_{e}\right)\right\}$ and $X=\;\left\{X_{1}\left(x_{g},y_{g}\right),X_{2}\left(x_{h},y_{h}\right),X_{3}\left(x_{k},y_{k}\right),X_{4}\left(x_{l},y_{l}\right)\right\}$. Similar to ICP, the centers of P and X are calculated by Equation (19).
(17)$$\begin{array}{c} u_{P}\left(x_{P},y_{P}\right)\;=\frac{1}{n}\overset{n}{\underset{1}{\sum }}P_{i}=\frac{1}{4}\left(x_{b}+x_{c}+x_{d}+x_{e},y_{b}+y_{c}+y_{d}+y_{e}\right)\\ u_{X}\left(x_{X},y_{X}\right)\;=\frac{1}{n}\overset{n}{\underset{1}{\sum }}X_{i}=\frac{1}{4}\left(x_{g}+x_{h}+x_{k}+x_{l},y_{g}+y_{h}+y_{k}+y_{l}\right) \end{array}$$

Then, the covariance matrices of P and X are calculated by Equation (20).
(18)$$W_{PX}=\overset{n}{\underset{1}{\sum }}\left(P_{i}-u_{P}\right){\left(X_{i}-u_{X}\right)}^{T}$$

Covariance matrix $W_{PX}$ by SVD is decomposed as shown in Equation (21).
(19)$$W_{PX}=U\left[\begin{array}{ccc} \varepsilon_{1} & \cdots & 0\\ \vdots & \ddots & \vdots \\ 0 & \cdots & \varepsilon_{n} \end{array}\right]V^{T}$$
where $\varepsilon_{1}\cdots \varepsilon_{n}$ are the singular values of $W_{PX}$ while the columns of matrix U are the eigenvectors of $W_{PX}W_{PX}{}^{T}$, and the columns of matrix V are the eigenvectors of $W_{PX}{}^{T}W_{PX}$. Then, the rotation matrix $R^{\prime }$ and translation matrix $t^{\prime }$ can be obtained by Equation (22).
(20)$$\begin{array}{c} R^{\prime }=VU^{T}\\ t^{\prime }=u_{x}-R^{\prime }u_{p} \end{array}$$

The SVD process in reflector positioning is the same as for ICP in Section 2.1. Here, *R*′ is the rotation matrix that makes the XOY coordinate system in Figure 13 rotate to the $X^{\prime }O^{\prime }Y^{\prime }$ coordinate system, and $t^{\prime }$ is the translation matrix from point O to point $O^{\prime }$ in Figure 13.

#### 2.2.3. Optimization of Improved Reflector Positioning Method

The above steps for calculating pose transformation are based on at least three reflectors being detected by LiDAR. However, the number of detected reflectors is possibly less than three. Figure 14a shows that when the number of detected reflectors is one or two, the corner points or edge points calculated in Section 3 are added to the initial reflector sets of two consecutive frames. If no corner points or edge points are detected, the two points of the wall that have a certain distance from the reflector are added to the set. As shown in Figure 14b,c, the red points near reflector a in the XOY coordinate system and reflector $a^{\prime }$ in the $X^{\prime }O^{\prime }Y^{\prime }$ coordinate system are corner points near reflector A in the global coordinate system. The red points near reflector b in the XOY coordinate system and reflector $b^{\prime }$ in the $X^{\prime }O^{\prime }Y^{\prime }$ coordinate system are the points of the wall to which reflector B is close in the global coordinate system. Then, with such red points added to the point pair sets, the pose transformation is calculated by SVD.

For the case without reflector detected, the transformation equals the last calculated transformation. Based on such optimization, the improved reflector positioning method can calculate a pose transformation between two consecutive positions in any conditions.

#### 2.2.4. Integrate the Pose Transformation into ICP

After obtaining a more precise initial iteration value, the ICP algorithm is operated to calculate the final pose transformation between two consecutive point cloud frames as described in Section 2.1.2. With the initial iteration value, the point clouds of the first frame are transformed to a new first frame according to the rotation and translation matrices. Then, in the second frame, the points closest to the points in the new first frame are found. These closest point pairs are used to calculate a new pose transformation. Repeat this process until the error function in Equation (3) is less than a threshold or the number of iterations reaches a threshold. After the ICP algorithm, the final pose transformation between two consecutive frames is obtained. The final LiDAR position is calculated by accumulating these pose transformations.

According to the content introduced above, the specific implementation steps of the LiDAR positioning algorithm based on ICP and artificial landmark assistance proposed in this paper are summarized as follows:Step 1: Obtain consecutive point cloud frames from the LiDARStep 2: Extract the coordinates of reflectors in frame i + 1Step 3: If the number of reflectors is more than three, construct a triangle set; if not, add corner points and so on to the reflector sets before constructing the triangle setStep 4: Match the triangle sets of frame i and frame i + 1 and obtain the correspondence of reflectors between two framesStep 5: Calculate the initial pose transformation by SVDStep 6: Provide the initial pose transformation to the initial iteration value of ICP and operate ICP to calculate the final pose transformationStep 7: Add the final pose transformation to the last position of LiDARStep 8: Return to step 1, repeat the above steps.

The specific algorithm flows are shown in Algorithm 1.


**Algorithm 1.** Improved least square calculating coordinate of reflectorInput:PCL:The original point clouds obtained from the LiDAROutput:P_LiDAR:Position of LiDAR1: Initialize:i ← 1,n ← 0,P_LiDAR [0] ← 02: while true do3:   Detect the reflectors and calculate the coordinates from PCL[i], structure the set of reflectors, n ← number of reflectors4:   if n ≥ 15:     if n > 26:       Structure the triangle set Tri7:     if n ≤ 28:       Add edge points, corner points or points of wall to R9:       Structure the triangle set Tri10:     Compare current set Tri and last set Tri′ and Find the correspondence of reflectors11:     Calculate the initial transformation, Rotation matrix $R_{i}$ and translation matrix $t_{i}$
12:  else 13:     $R_{i}=R_{i-1}$, $t_{i}=t_{i-1}$
14:  end if15:  Provide $R_{i}$ and $t_{i}$ to the initial transformation of iteration16:  Use ICP algorithm to calculate final transformation $R_{i}^{\prime }$ and $t_{i}^{\prime }$
17:  Output the position of LiDAR: P_LiDAR[i] ← P_LiDAR[i − 1] + ($R_{i}^{\prime }$,$t_{i}^{\prime }$)18:   i ← i + 1, n ← 019: end while


## 3. Results

### 3.1. Platform Construction and Data Collection

The platform and environment of the experiment are shown in Figure 15. The platform sensors were an NAV350 LiDAR (SICK company, Waldkirch, Germany) and an LR-1F LiDAR (OLEI company, Hangzhou, China). They are both 2D LiDARs. NAV350′s scanning frequency is 8 Hz, its scanning range is 360°, with a resolution of 0.25°. It can scan reflectors within 70 m with a precision of ±4 mm. LR-1F’s scanning frequency is 10~25 Hz. In this study, its frequency was 10 Hz and its scanning range was 360°, with a resolution of 0.25°. It can scan reflectors within 50 m. Windows system software provided by SICK company was used to collect and process the NAV350 data. The LR-1F data were processed in the Ubuntu16.04 system. The experimental environment was an L-shaped underground garage about 50 m long and 5 m wide. It was changed by using the reflectors, as shown in Figure 4. The reflectors were randomly suspended on the wall of the underground garage with a height of 1.7 m. NAV350 costs RMB 55,000 and LR-1F costs RMB 7000.

The height of LR-1F was set to 2 m and NAV350 was slightly lower than that of LR-1F, preventing NAV350 from disturbing the LR-1F scanning. The experiment was divided into two parts. The first part verified the precision of the reflector coordinate fitting algorithm and the second part compared the positioning performance of NAV350 with that of LR-1F. NAV350 uses a reflector positioning algorithm whose positioning precision can reach 4 mm. Therefore, it was used as the ground truth in the experiment. During the experiment, the NAV350 Windows system software received the data and exported the positioning data. The positioning data storage format is shown in Table 1. The LR-1F data were processed by the Robot Operating System (ROS) in the Ubuntu system, and the corresponding positioning data were exported as TXT files for subsequent processing.

### 3.2. Experiment of Fitting the Reflector Coordinates

The experiment was conducted in the environment shown in Figure 16a. The placement of reflectors simulated the actual situation in which a reflector is close to a flat wall.

For the experiment, we used two different LiDARs, so the coordinate systems were different. It is difficult to compare the coordinates of different coordinate systems. The most direct method is to convert all coordinates to the same coordinate system according to the transformation between coordinate systems. However, the transformation between coordinate systems of two different LiDARs is impossible to accurately calculate or measure. Therefore, it was impossible to directly compare coordinates, but the distance between reflectors is constant in both coordinate systems. Therefore, the distance between reflectors was chosen as the object for comparison.

The fitting performance of NAV350 (shown in Figure 16b) is presented in Figure 17. The coordinates of three reflectors are $\left(0.023\;m,-1.006\;m\right)$, $\left(-0.924\;m,0.212\;m\right)$, and $\left(-0.076\;m,0.808\;m\right)$. By calculating, the distances of the three reflector pairs were found to be 1.5428, 1.0365, and 1.8167 m, respectively. Then, as shown in Figure 16c, we used LR-1F to verify the traditional least square method and the algorithm proposed in Section 3. Finally, we compared the lengths calculated by different algorithms and the errors are shown in Figure 17 and Figure 18.

Figure 17 shows the lengths calculated by the three algorithms. Under the premise of using the NAV350 measurements as ground truth, Figure 18 shows the errors of the least square and the proposed methods.

### 3.3. Experiment with the Proposed Positioning Algorithm

Before the experiment with the proposed positioning algorithm, NAV350 was used to map the experimental environment in advance. NAV350 was connected to the Windows system by an RJ-45 port. The software started in mapping mode and then the platform was moved in the environment. LiDAR moved with the experimental platform. When NAV350 moved, we had to ensure NAV350 could scan three reflectors of the constructed map. In addition to the reflectors of the constructed map, if new reflectors were detected, their coordinates were calculated and added to the map. When all the reflectors were included in the map, the global map for NAV350 was successfully constructed. 

From the results of the experiment with the positioning algorithm, we compared the performance of the reflector positioning system (NAV350), PL-ICP [24] algorithm, and the proposed method. The PL-ICP algorithm was implemented using the laser-scan-matcher package in ROS. The final trajectories are shown in Figure 19.

In terms of positioning precision, the error is difficult to calculate. Because the frequencies of two LiDARs are different, the error cannot be calculated point-to-point. Thus, we chose the two points in the NAV350 trajectory closest to the point in the trajectory of the proposed method. Then, we calculated the distance between the point and the line formed by the two closest points. These distances can represent the error of two trajectories. The error was calculated in the road sections where the two trajectories were coincident. The result of error is shown in Figure 20.

## 4. Discussion

### 4.1. Performance of Fitting the Reflector Coordinates

Table 2 generalizes Figure 17 and Figure 18. Figure 17 shows that the distances between reflectors calculated by the proposed method are closer to the ground truth whose averages are 1.0367, 1.5450, and 1.8170 m. In comparison, the average distances calculated by the least squares method are 1.0275, 1.5235, and 1.7898 m, respectively. In the error analysis in Figure 18, the error of the proposed method floats around zero. The average errors are $1.6380\times 10^{-4}$ m, 0.0022 m, and $3.3292\times 10^{-4}$ m, respectively, which reach millimeter-level precision. However, the average errors of the least square method are −0.0009, −0.0193, and −0.0269 m, respectively.

Figure 18 and Table 2 show that the coordinates of the reflectors fitted by the proposed method are more precise than those of the traditional least square method. Additionally, the precision of the proposed method does not change with increasing distance of the reflector. This means the proposed method is more stable. In addition, there is no iteration in the calculation process, which ensures real-time performance and reduces the computing resources requirements. The higher fitting precision reduces the possibility of mismatching in the subsequent positioning to further improve the positioning precision.

### 4.2. Performance of the Proposed Positioning Algorithm

Overall, the trajectory of the proposed algorithm almost coincides with the unmissed part of the NAV350 trajectory, and the trajectory of the PL-ICP algorithm is shorter. This is because the experimental environment has few natural features in some road sections, creating the point clouds matching error for the PL-ICP algorithm, and the calculated pose transformation between frames is very small. After such errors gradually accumulate, the trajectory shortens, indicating that the PL-ICP algorithm is unsuitable for environments with few natural features. The proposed algorithm maintains better positioning performance even in such environments, which compensates for the shortcomings of the PL-ICP algorithm. In addition, the parts of the trajectory marked by dotted lines in Figure 19 show that sections of the NAV350 trajectory are missing. Next, the reasons for the different missing trajectory sections and the positioning error of the algorithm are analyzed in detail.

The missing sections of the trajectory of NAV350 are shown in Figure 21a–d.

There are three different types of missing sections. The first kind is shown in Figure 21a and the lower missing section of Figure 21b. This kind of missing trajectory is due to the calculation latency of NAV350, as shown in Figure 21e,f. The red and green points are the detected reflectors, and the hollow points are the reflectors in the constructed global map. In Figure 21f, the detected reflectors do not perfectly match the corresponding ones in the global map. The reflector matching process is delayed or unable to match successfully for some reason, resulting in delays and sudden changes in positioning data and missing the trajectory. The second kind, as shown in the missing upper section in Figure 21b, is due to the mismatch of reflectors, as shown in Figure 21g,h. The correct match of the three detected reflectors is shown in Figure 21g. In Figure 21h, the same three detected reflectors match the other three reflectors in the global map. Usually, a set of detected reflectors only has one corresponding set of reflectors in the global map. This mismatch condition occurs when there are similar triangles in the global map, which is difficult to avoid in a large environment that needs many reflectors. The positions calculated under the condition in Figure 21h are shown in Figure 21d. The upper red outlier points are the positioning results in Figure 21h. These points should appear in the missing upper section of the trajectory in Figure 21b, but due to the mismatch, they are nearly 20 m away from where they should be. The last kind of missing trajectory is shown in Figure 21c, resulting from the number of detected reflectors being less than three. In the environment, no reflectors were placed along the road section. Therefore, NAV350, which is based on the reflector positioning system, could not calculate the position and no positioning data were exported, leading to part of the trajectory being missed.

In summary, the stability of NAV350, namely the reflector positioning system, is considerably limited by the layout of the reflectors. In comparison, the method proposed in this article does not experience this problem. The blue trajectory in Figure 19 shows that the proposed method performs well in road sections where NAV350 cannot calculate position. The trajectory of the proposed method is continuous without missing parts or outliers. This indicates that the proposed method does not experience problems such as latency or mismatch. Even in the road section without reflectors, it also maintained good real-time and robust performance. 

In Figure 20, the maximum error is 0.2925 m, the average error is 0.0483 m, and most error is under 0.1 m. The error at the distance of 40 to 50 m suddenly increases because the experimental platform rotates at the position shown in Figure 22 with LR-1F as the center. Therefore, during the rotation, the displacement of LR-1F is small. The error at the distance of 50 to 60 m also increases for this reason. The error at the distance of 90 to 100 m increases due to the certain distance between two LiDARs. The error at the end of the route represents the certain distance between the two LiDARs. Limited by the experimental platform, the calculated error is larger than the actual error. In actuality, the average error is less than 0.0483 m.

In summary, the proposed method is more stable and robust than the traditional reflector positioning system and ICP algorithm, and is cheaper. The precision of reflector positioning system proposed by Xu et al. [15] is better than 0.08 m with the AGV route known in advance. The positioning method proposed in this paper is more precise and has an error less than 0.0483 m without the limit of knowing the route in advance. Additionally, compared to positioning systems based on SLAM such as that constructed by Cho et al. [21], the method in this paper does not need to store a map in advance. Compared with the 0.036 m precision of the method proposed by Cho et al., our method’s precision is acceptable.

## 5. Conclusions

This paper first introduced the principle of traditional reflector positioning and ICP algorithms and then we improved upon the traditional ICP algorithm. The initial value calculated by reflector positioning was provided for iterative calculation, and a LiDAR positioning method based on the ICP algorithm and reflectors assistance was proposed. This method is not limited to environments with many natural features: It can be applied to sparse environments such as warehouses or corridors. It combines artificial landmarks, namely reflectors with natural features, which improves its robustness. In the latter part of the paper, the proposed method was compared with the existing reflector positioning system applied to production. The proposed method maintains accurate positioning performance in the section where the reflector positioning system cannot calculate position or calculates a wrong position. The experimental results showed that compared with the traditional ICP algorithm and the existing reflector positioning system, the proposed algorithm can reach centimeter-level precision and is cheaper, with an average positioning error of less than 0.05 m and higher robustness. However, this method still belongs to the cumulative positioning algorithms. Therefore, in future research, we will consider adding a global positioning algorithm that can further improve precision and robustness.

## Figures and Tables

**Figure 1 sensors-21-07141-f001:**
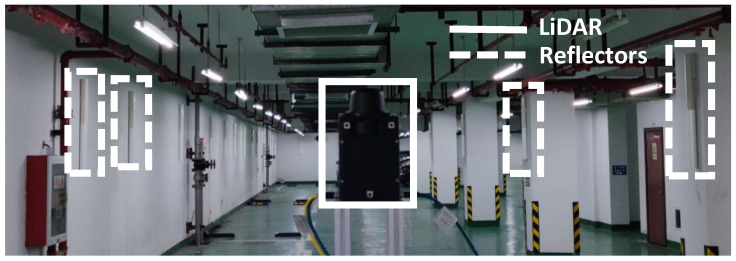
Hardware required for the reflector positioning system. The black device indicated by the solid line is a 2D LiDAR. The white pipes indicated by dotted lines are reflectors.

**Figure 2 sensors-21-07141-f002:**
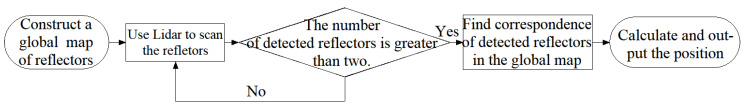
Flow chart of the traditional reflector positioning algorithm.

**Figure 3 sensors-21-07141-f003:**
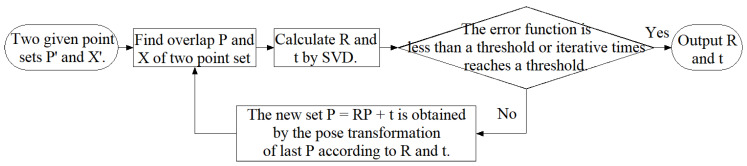
Flow chart of the ICP algorithm.

**Figure 4 sensors-21-07141-f004:**
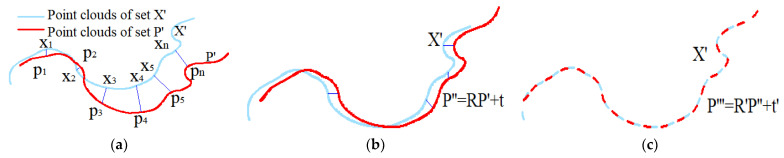
Schematic diagram of the ICP algorithm. (**a**) The closest point $x_{n}$ of point $p_{n}$ is regarded as the corresponding point of point $p_{n}$. (**b**) Calculate the pose transformation and find new correspondences. (**c**) Calculate new pose transformation based on new correspondences.

**Figure 5 sensors-21-07141-f005:**
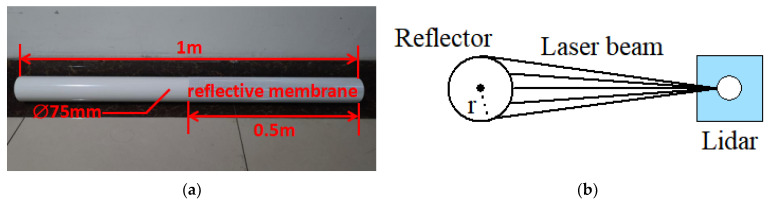
Details about reflectors. (**a**) An example of a reflector in this paper. The reflector is a cylinder with a length of 1 m and a diameter of 75 mm covered by a reflective membrane (**b**) Standard of selecting dimension of reflectors. This standard ensures that the LiDAR can scan enough point clouds to fit the reflector coordinates.

**Figure 6 sensors-21-07141-f006:**
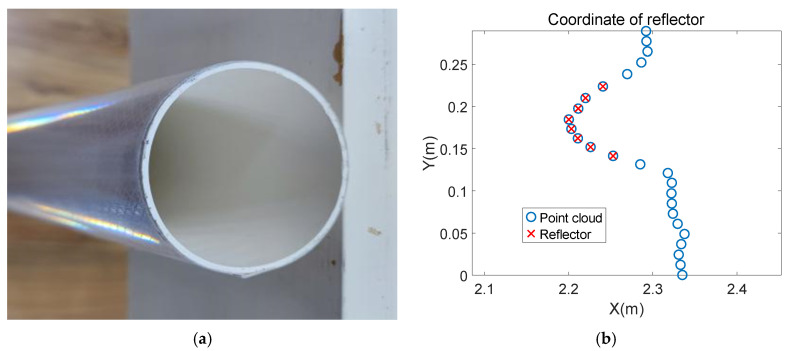
LiDAR scans the reflector placed close to a wall. (**a**) A vertical view of a reflector placed close to a wall. The cross section of the reflector is a circle and the cross section of the wall is a straight line. It can be seen from the placement that the circle is close to the line. (**b**) Schematic diagram of point clouds obtained by LiDAR scanning (**a**).

**Figure 7 sensors-21-07141-f007:**
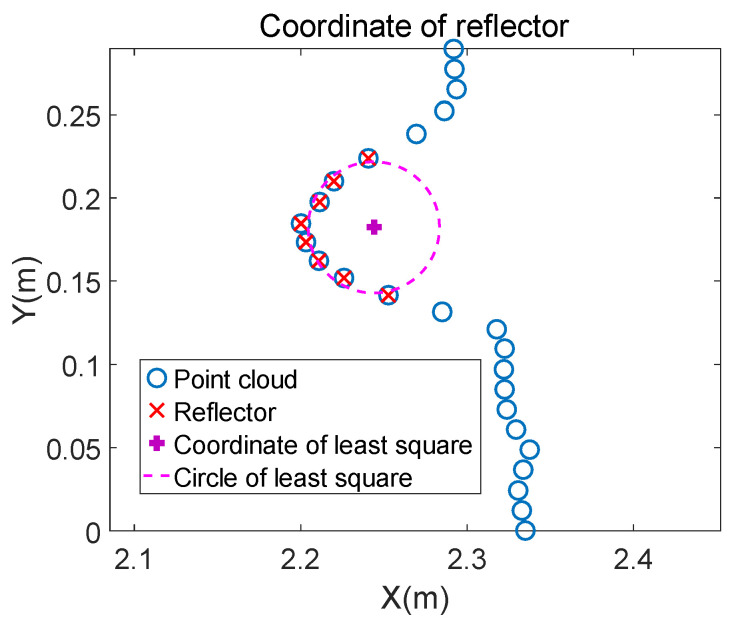
The reflector coordinate is fitted by the traditional least square method firstly. The fitted circle is still a certain distance from the line. It is necessary to modify the rough coordinate to get a more precise value.

**Figure 8 sensors-21-07141-f008:**
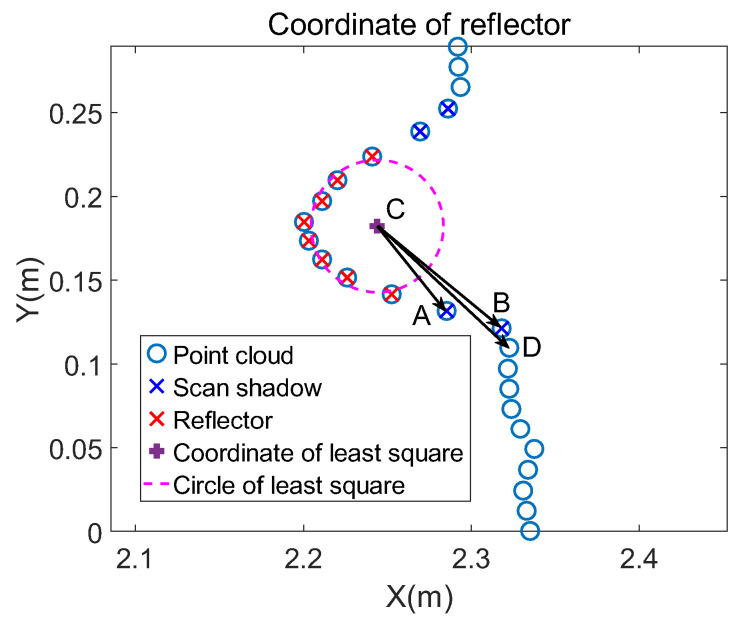
The blue points are scan shadows of the reflector. The reflector is placed in front of the wall. Therefore, a laser beam is divided into two beams. One radiates on the edge of the reflector and the other one radiates on the wall. This makes the final point cloud between the reflector and wall and affects modifying the coordinate.

**Figure 9 sensors-21-07141-f009:**
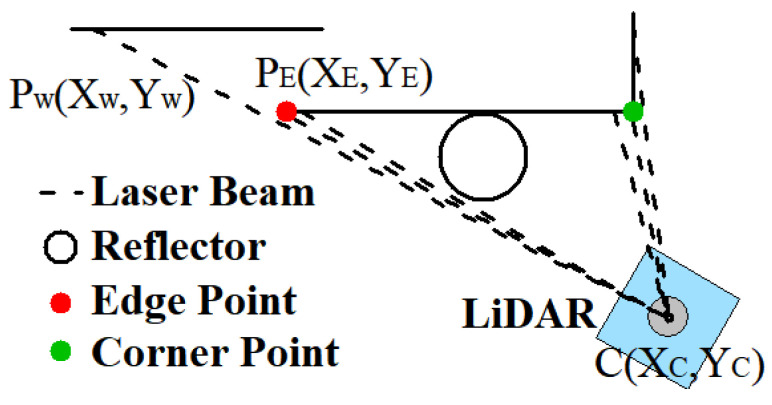
Example of two kinds of boundaries. The red one is called edge point. This type exists at the edge of the wall. The green one is called corner point, which represents the corner of the wall.

**Figure 10 sensors-21-07141-f010:**
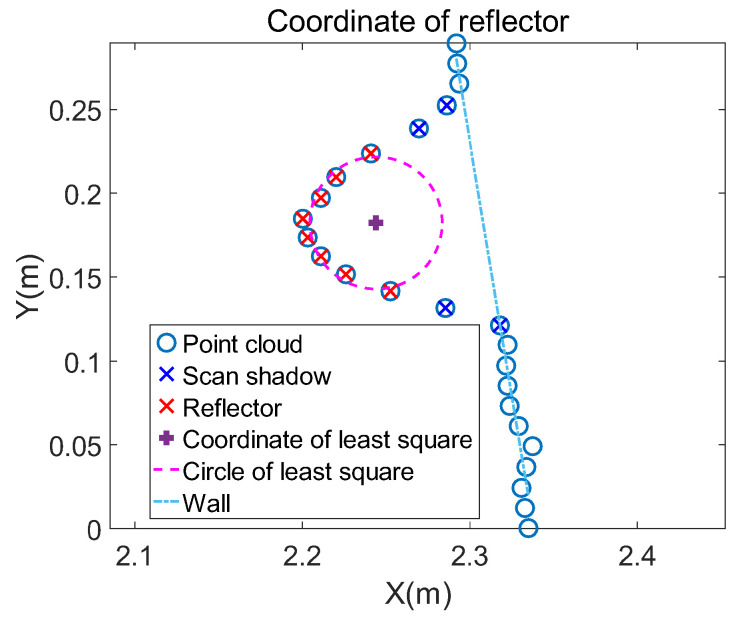
After removing scan shadows, the remaining point clouds belongs to the wall except reflector point clouds. The two boundaries determine the wall range. The line of the wall cross section can be fitted according to point clouds within the range.

**Figure 11 sensors-21-07141-f011:**
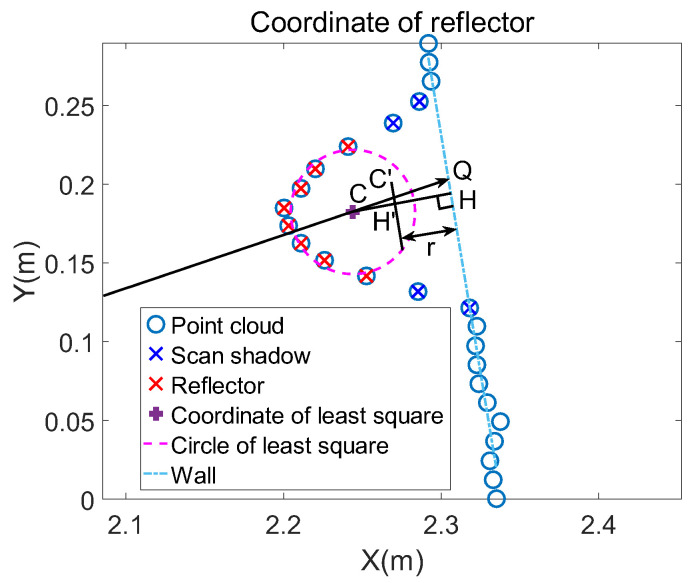
The reflector is placed close to the wall. The distance between the circle center and the wall equals the radius. Therefore, move the rough circle center to a point whose distance from the wall is radius.

**Figure 12 sensors-21-07141-f012:**
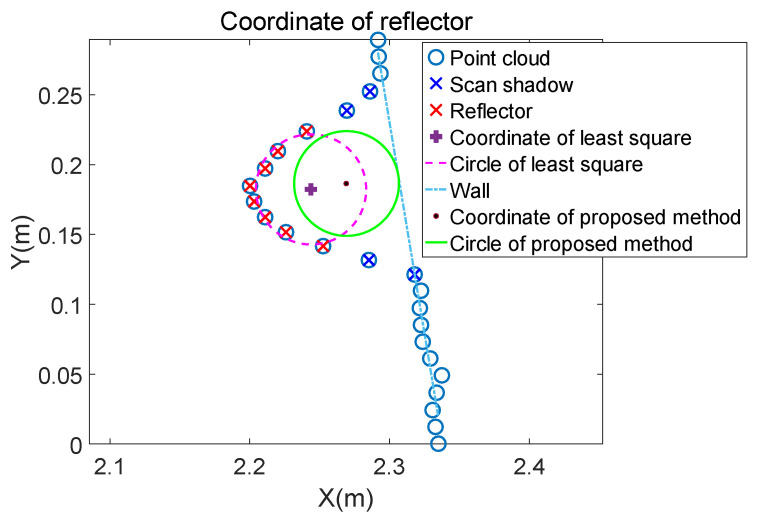
After modification, the final reflector coordinate is more precise than the initial one fitted by the traditional least square.

**Figure 13 sensors-21-07141-f013:**
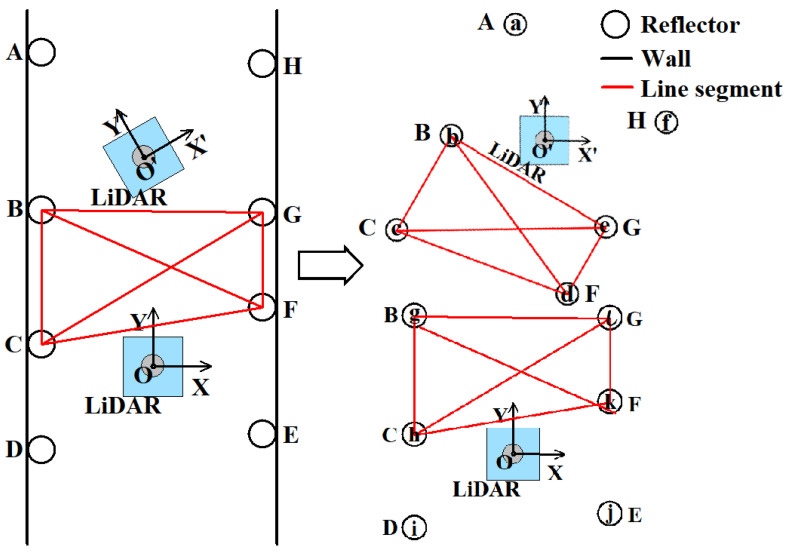
The LiDAR scans reflectors in two positions. Four same reflectors are scanned in both two point cloud frames. Therefore, if correspondences of four reflectors in two frames are known, the pose transformation can be calculated based on their coordinates.

**Figure 14 sensors-21-07141-f014:**
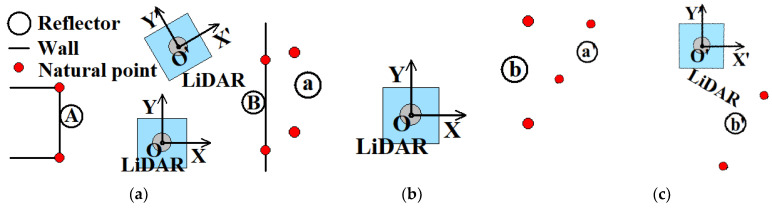
Schematic diagram of the reflector positioning method with one or two reflectors. (**a**) Add natural feature points to the initial set when detected reflectors are insufficient. (**b**) Scanning results of the first frame after natural points are added. (**c**) Scanning results of the second frame after natural points are added.

**Figure 15 sensors-21-07141-f015:**
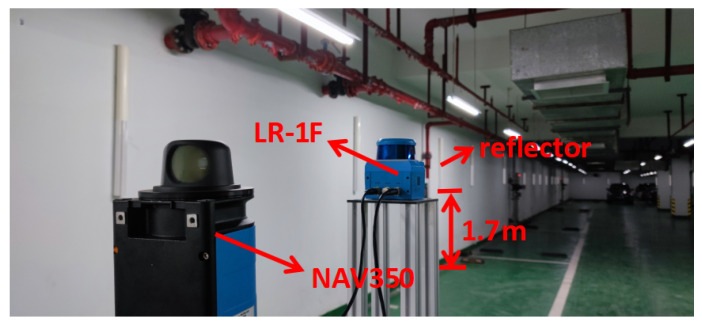
The platform and environment of the experiment. The height of LR-1F was set to 2 m and NAV350 was slightly lower than that of LR-1F. The experimental environment was an L-shaped underground garage about 50 m long and 5 m wide. The reflectors were randomly suspended on the wall with a height of 1.7 m.

**Figure 16 sensors-21-07141-f016:**
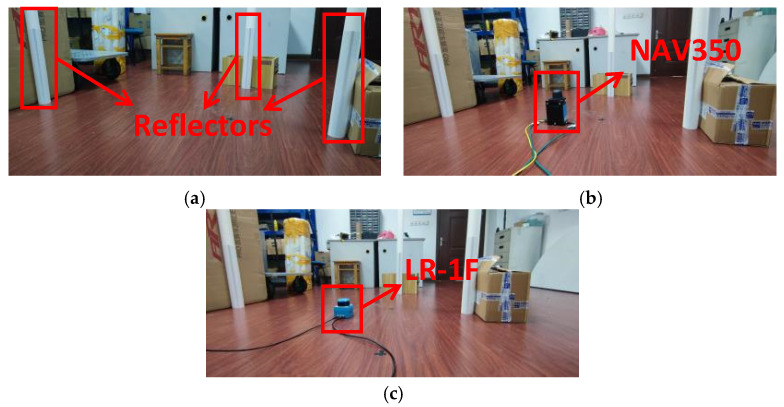
The experiment of fitting the coordinates of the reflectors. (**a**) Reflectors were placed close to flat planes. This simulated placement of reflectors in real scenes. (**b**) NAV350 scanned the reflectors and calculated distances between reflectors. (**c**) LR-1F scanned the reflectors and calculated distances between reflectors.

**Figure 17 sensors-21-07141-f017:**
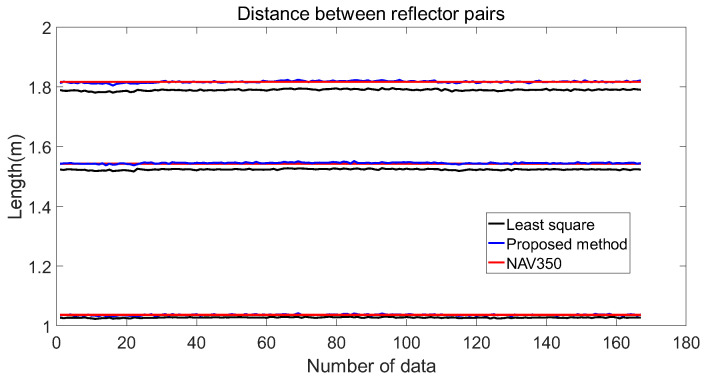
The distances between reflector pairs calculated by different methods. Black lines are the distances calculated by the traditional least square. Blue lines are the distances calculated by the proposed method. Red lines are the distances calculated by NAV350, which uses a traditional reflector positioning system. Blue lines almost coincide with red lines and the black lines do not.

**Figure 18 sensors-21-07141-f018:**
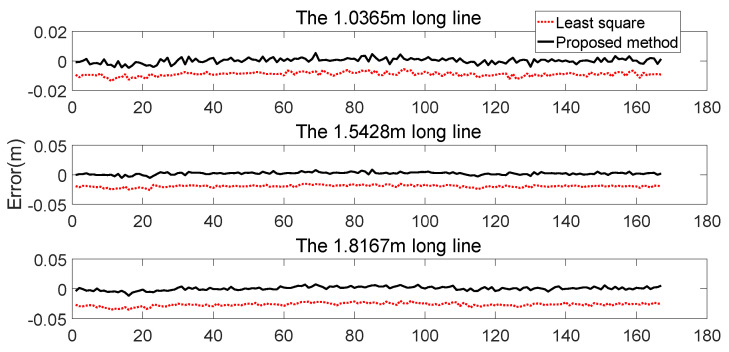
Errors of the traditional least square and proposed methods using NAV350 as ground truth. It can be seen that the error of the proposed method is always less than that of the traditional least square.

**Figure 19 sensors-21-07141-f019:**
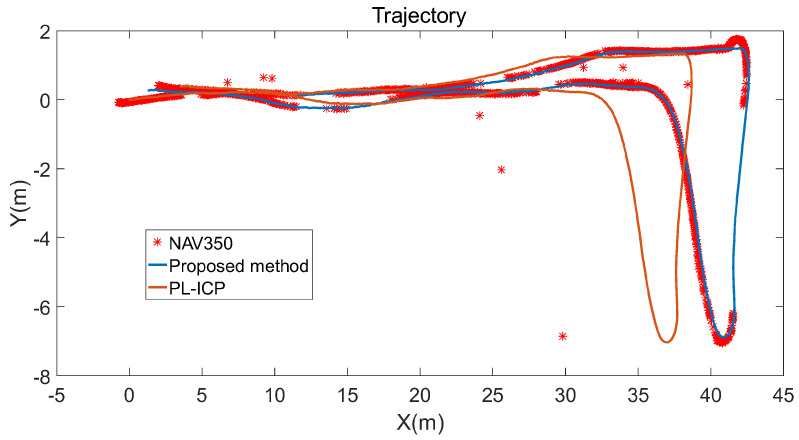
Trajectories of the three positioning algorithms. Red points are the positions calculated by NAV350. The blue line is the trajectory calculated by the proposed method. The brown line is the trajectory calculated by the traditional least square.

**Figure 20 sensors-21-07141-f020:**
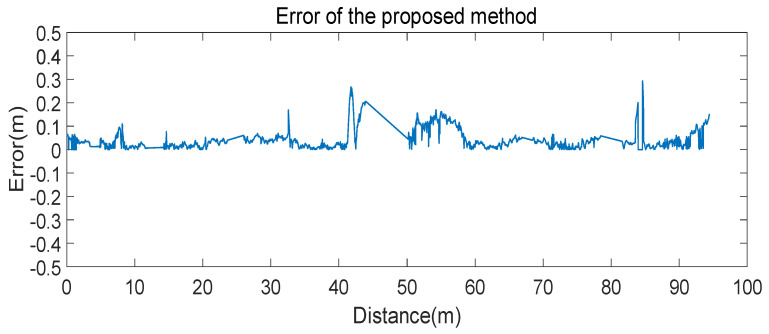
The blue line indicates the errors in different positions. Due to missing sections of NAV350 trajectory, there are sudden changes near several errors. Most errors are less than 0.1 m.

**Figure 21 sensors-21-07141-f021:**
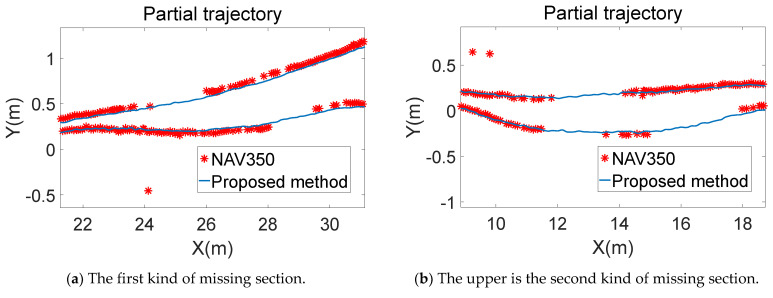
Analysis of missing sections of the NAV350 trajectory. There are three types of missing sections. The first type is the missing sections in (**a**) and lower missing section in (**b**). The cause of this type is latency in (**e**,**f**). The second type is the upper missing section in (**b**) which is caused by mismatch in (**g**,**h**). Positions calculated by mismatching are in (**d**). (**c**) is the third one caused by insufficient reflectors.

**Figure 22 sensors-21-07141-f022:**
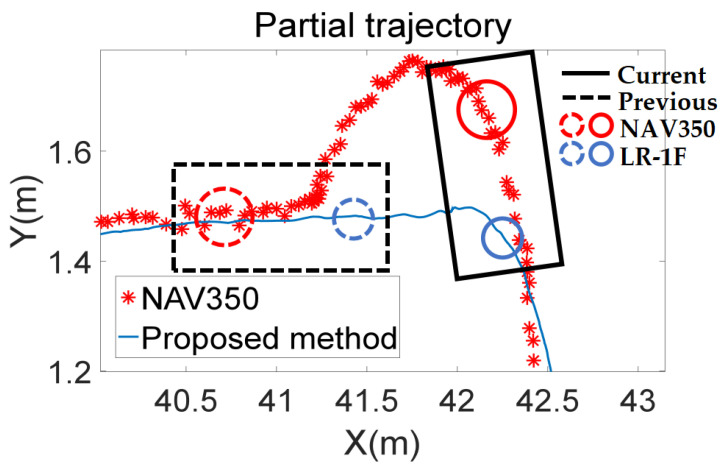
Schematic diagram of the reason for sudden increasing error. The rectangles indicate the positions of the platform at different time stamps. The circles indicate the positions of LiDARs at different time stamps. Dotted lines indicate the previous position and solid lines indicate the current position. The platform rotates with LR-1F as the center during two time stamps.

**Table 1 sensors-21-07141-t001:** Data storage format of NAV350.

Frame_id	X (m)	Y (m)	Angle (°)
1	−0.647	0.224	356.46
2	−0.647	0.225	356.46
3	−0.646	0.225	356.46
4	−0.646	0.228	356.49
5	−0.648	0.224	356.46
6	−0.648	0.226	356.47
7	−0.648	0.227	356.48
8	−0.646	0.222	356.44
9	−0.647	0.224	356.45
10	−0.646	0.223	356.45
11	−0.649	0.225	356.45
12	−0.648	0.225	356.46
13	−0.647	0.224	356.48
14	−0.647	0.224	356.51
15	−0.645	0.224	356.51
…	…	…	…

**Table 2 sensors-21-07141-t002:** Different algorithms’ performance of fitting reflector coordinates.

	NAV350	Method Proposed	Least Square
Average length of the 1st line (m)	1.0365	1.0367	1.0275
Average length of the 2nd line (m)	1.5428	1.5450	1.5235
Average length of the 3rd line (m)	1.8167	1.8170	1.7898
Average error of the 1st line (m)	/	$1.6380\times 10^{-4}$	−0.0090
Average error of the 2nd line (m)	/	0.0022	−0.0193
Average error of the 3rd line (m)	/	$3.3292\times 10^{-4}$	−0.0269

## Data Availability

The data presented in this study are available in Section 3 and Section 4.

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
