# Peer review of "LiDAR Positioning Algorithm Based on ICP and Artificial Landmarks Assistance"

_sensors, 2021, doi:10.3390/s21217141_

Round 1
Reviewer 1 Report
This paper proposes a Lidar positioning algorithm based on ICP and artificial landmarks assistance. Authors claim it enables the positioning algorithm to maintain a certain precision when landmarks detection is disturbed. Also, they claim that the results obtained by the proposed method, when compared with the positioning scheme developed by DICK, shows that it can effectively improve the positioning precision.
First, it is not common to use references in the abstract. Please remove.
Acronyms must be defined/explained before first use.
English must improve. Text is very difficult to understand. Many typos and grammatically incorrect phrases.
The way the authors refer to other works is strange: “reference [10]” instead of “Ronzoni et al. [10]”.
The text is difficult to understand, the paper structure is not adequate, and it does not reach the standards of a scientific publication. It must be improved to be considered for publication.
Reviewer 2 Report
This paper presents the improved version of the Iterative Closest Point (ICP) algorithm for enhanced performance using artificial landmark assistance.
Major comments
While this reviewer thinks that the overall idea and approach have merit to be published due to its better robustness and accuracy, the presentation has a lot to be desired. The presentation has a significant impact on impression. This reviewer was very excited in the beginning reading the abstract, but it diminished rapidly as I started to read the manuscript. A major overhaul should be made for this manuscript to be considered publishable.
- This reviewer recommends shortening the 2.1 traditional Reflectors Positioning system significantly as it is not the method this paper uses. The algorithm and relevant formulas are simple enough to be explained in words. If authors believe this has to stay, perhaps make the manuscript more coherent such that the readers can think that this is necessary.
- Make it clear up to which point is the literature review so that readers can understand the contribution of this paper more clearly. Evidently, chapter 2 is a literature review. Yet. It is not clear the role of the least squares method presented in chapter 3. Typical papers have the following flow: 1) introduction (with literature review) 2) materials and methods 3) results 4) discussion 5) conclusion.
- There are two sections 2.1s.
- Section 2.1 starts with the full name of ICP is… The section could have started with more attractive or valuable information, such as why and how ICP emerged onto the AGV scene.
- Figure 1: try adding legends of the equipment within this figure. Every figure should be self-explanatory.
- Figure 2: the purpose of this figure is not clear. Is there any connection between this and figure 1? Are the reflectors shown in figure 2 from figure 1? It appears that this figure is just a snippet of a complete graph, and it does not look professional. Be a little more descriptive with figure captions.
- Figure 4: This reviewer suggests removing the coordinates from the figure since it will make the figure look less cluttered.
- Figure 5: This flowchart looks dated with inconsistent languages (Chinese?) and low resolution. Please make it look more contemporary.
- Figure 6: If you have subfigures within a figure like this one, i.e., (a), (b), (c). Readers expect the captions to explain what each figure is for.
- All graphs should be provided using vector graphics. Graphs in figure 3 are not bad, but figures 2, 4, 6, 8b, 9, 10, etc., could be vastly improved.
- The presentation of the formulas could be better. The fonts and sizes of the formulas within the text are inconsistent. The way fractions are shown is not consistent.
- Formula 17&18 do not look like a formula.
- Formulas are repeated. See formulas 6 and 21.
- Figure 18, just by looking at this figure, it is difficult to understand how the experiment is conducted.
- Figure 19, it is not common to use the screenshot to show the settings of software like this.
Minor comments
- Be consistent with abbreviations. There are abbreviations that are missing the definition. For instance, ICP (Iterative Closest Point) is not defined at the beginning of the manuscript. Instead, it is defined on page 5. LOAM (Lidar Odometry and…).
- Have a space before putting references
- reference[10], reference[8], etc. – this is not a usual way of referring to the literature. Rewrite. e.g., XX was discussed in [10]. Or XX et al. [10] presented XYZ.
- Page 2, Line 57: This paper discusses… What do you mean by ‘This’? Reference 10 or your manuscript?
- Page 2 line 73: This will affects – will affect
- Page 2, line 80: However, the article ‘only’ conducted… Do not use words that may undervalue other’s work. Tone it down. Authors of that literature may get offended.
- Page 3, line 104: SLAM has already been defined.
- Page 4, line 170: burden is not a scientific word. Try using quantitative words like time.
- Page 4 line 194: may has -> may have
These minor comments only show the errors in the first several papers. Authors should carefully proofread the manuscript to catch all mistakes.
Reviewer 3 Report
Dear Authors
Your article: “Dealing Positioning Algorithm Based on ICP and Artificial Landmarks Assistance” is very interesting and can contribute to the dissemination of the use of LiDAR sensors in the bi/three-dimensional positioning of robots. After a thorough reading of the article, I presented 32 comments indicated in the digital file and I present below my contributions to the improvement of the final version of the article:
1) Line 155 to 166: review and improve the writing.
2) Insert a legend for Figure 6 referring to the red and blue lines.
3) Line 215: “transformation between consecutive point cloud frames is not too large”. What did you mean by is not too large?
4) Line 247: “A reflector is a cylinder with a length 247 of 1m and a diameter of 7.5cm”. Are these the standard dimensions for a reflector? Is the default cylindrical shape? Comment more about choosing this reflector pattern.
5) In Figure 8b, are the points cloud the points referring to the wall? It would be interesting for you to rotate image 8a (90° clockwise) as shown in figure 8b. Replace in legend: point cloud to the wall points.
6) Present the nomenclature used in formulas 8 and 9.
7) Formulas 17 and 18: Please check the nomenclature and explain formula development.
8) Lines 414 to 438: Revise the text and nomenclature used. The term Pairs is used in two different contexts and this can confuse the reader.
9) Line 499: improve the characterization of the LiDAR sensors used: angular and linear precision. Was an IMU/compass used with the sensors? If yes, please detail. Detail the city and place (dimensions) where the experiments were carried out.
10) You commented that you would use data from an IMU/compass to improve the performance of the algorithm as mentioned in chapter 1. This issue was not highlighted in chapters 3 and 4.
11) Line 513: How the 4 mm threshold was determined.
12) Add the unit of measure in Table 2.
13) In chapter 5 it would be interesting to compare the results obtained in the study developed by you with those carried out by the authors mentioned in the bibliographic references.
14) Line 563: "is moved in the environment". How was the movement/displacement of the sensor made along with the study site? Please detail.
15) Line 641: "with a cheaper cost." You didn't address the issue of sensor costs in the article. It would be interesting to mention the acquisition costs of each sensor.
16) Please quote Reference number 4 in writing English.
I conclude by congratulating them for the work they have done and for the article presented!
Best regards.
Round 2
Reviewer 1 Report
Figures and algorithms are out of limits and/or with poor quality. It is difficult to read the text inside the figure/algorithm. Dot missing line 178 before “However”. Sentences are not well written, some very short, other difficult to understand.
The text is still poorly written and difficult to understand. Unfortunately, It does not reach the standards of a scientific publication.
Author Response
1.Figures and algorithms are out of limits and/or with poor quality. It is difficult to read the text inside the figure/algorithm.
The size of figures, algorithms and the text inside them has been adjusted.
2.Dot missing line 178 before “However”. Sentences are not well written, some very short, other difficult to understand.The text is still poorly written and difficult to understand. Unfortunately, It does not reach the standards of a scientific publication.
The English writing of the manuscript has been checked and modified by English Editing Service of MDPI.
Reviewer 2 Report
The authors seem to have revised the manuscript based on previous comments.
However, details on visuals, figures, and texts within them could still be improved.
This reviewer recommends updating the figure captions. For instance, in figure 7, it says 'Fitted by traditional least square', be more descriptive. What has been fitted, what is the inference? Figures should be self-standing, and self-explanatory.
A similar thing is occurring in Figure 8, 'the blue points are scanshadows' scan shadows of what? In order for this figure to be self-standing, captions on ALL FIGURES should be completely overhauled.
Author Response
1.However, details on visuals, figures, and texts within them could still be improved.This reviewer recommends updating the figure captions. For instance, in figure 7, it says 'Fitted by traditional least square', be more descriptive. What has been fitted, what is the inference? Figures should be self-standing, and self-explanatory.A similar thing is occurring in Figure 8, 'the blue points are scanshadows' scan shadows of what? In order for this figure to be self-standing, captions on ALL FIGURES should be completely overhauled.
More detailed captions have been added to all figures.
Reviewer 3 Report
Dear Authors
The second version of his article: "Dealing Positioning Algorithm Based on ICP and Artificial Landmarks Assistance" enables the reader to fully understand the mathematical models, methodologies and results achieved.
Thank you for sending the article with all the suggestions indicated and also the cover letter answering my questions.
Congratulations on your work!
Author Response
Thank you for your congratulations and patient guidance
Round 3
Reviewer 1 Report
Figures and algorithms, as well as the text, have been improved.
Congratulations to the authors.